# Effects of Replacing Soybean Meal Protein with *Chlorella vulgaris* Powder on the Growth and Intestinal Health of Grass Carp (*Ctenopharyngodon idella*)

**DOI:** 10.3390/ani13142274

**Published:** 2023-07-12

**Authors:** Linlin Yang, Minglang Cai, Lei Zhong, Yong Shi, Shouqi Xie, Yi Hu, Junzhi Zhang

**Affiliations:** 1College of Animal Science and Technology, Hunan Agricultural University, Changsha 410128, China; yll1085544200@163.com; 2College of Fisheries, Hunan Agricultural University, Changsha 410128, China; cml950518@outlook.com (M.C.); zhonglei-5@163.com (L.Z.); shiyong@stu.hunau.edu.cn (Y.S.); 3State Key Laboratory of Freshwater Ecology and Biotechnology, Institute of Hydrobiology, The Chinese Academy of Sciences, Wuhan 430072, China; sqxie@ihb.ac.cn

**Keywords:** *Ctenopharyngodon idella*, *Chlorella vulgaris*, growth performance, oxidative stress, intestinal morphology, intestinal microflora

## Abstract

**Simple Summary:**

The development of novel protein sources plays an important role in improving the economic benefit of aquatic products. The *Chlorella vulgaris* (*C. vulgaris*) powder is a novel non-grain single-cell protein with a high reproductive rate, short growth cycle, strong environmental tolerance and easy artificial cultivation. In this experiment, grass carps (initial weight: 20.13 ± 0.09 g) were fed diets by replacing 0% (SM), 25% (X25), 50% (X50), 75% (X75) and 100% (X100) of SM with *C. vulgaris* for 8 weeks. In conclusion, the *C. vulgaris* powder replacement of 50% soybean meal was recommended as feed for grass carp. However, the positive effects were apparently weakened when the soybean meal was replaced with high levels of *C. vulgaris* powder.

**Abstract:**

*Chlorella vulgaris* (*C. vulgaris*) powder is a novel non-grain single-cell protein with enormous potential to be a protein source. However, it is poorly studied in aquatic animals. The purpose of the present study was to explore the optimum replacement ratio of *C. vulgaris* powder and the influence of the substitution of soybean meal with *C. vulgaris* on grass carp (*Ctenopharyngodon idella*) in terms of growth performance, intestinal integrity and the microbial community. Five isonitrogenous and isolipidic diets were formulated by replacing 0% (SM, containing 30% soybean meal), 25% (X25), 50% (X50), 75% (X75) and 100% (X100) soybean meal with *C. vulgaris.* The feeding trial period lasted 8 weeks. At the end of the experimental trial, the X50 group showed higher FW, WGR and PER than the SM group (*p* < 0.05). The feed conversion ratio (FCR) of the X50 group was significantly lower than that of the SM group (*p* < 0.05). The X50 group showed the highest value of the goblet cell number, intestinal amylase and trypsin activities when compared with the SM group (*p < 0.05*). Replacing 50% soybean meal with *C. vulgaris* improved the intestinal barrier integrity, as evidenced by upregulating *zo-1*, *zo-2* and *occluding* transcript (*p* < 0.05), and alleviated oxidative stress by an increased SOD enzymatic activity and transcript level, probably mediated through the *Nrf2*-*keap1* signaling pathway (*p* < 0.05). Meanwhile, the X50 group enhanced intestinal immunity, as manifested by increased ACP and LZM activities (*p* < 0.05), and downregulated the *tlr-4*, *tlr-7*, *tlr-8* and *il-6* through the *tlr* pathway (*p* < 0.05). The functionally predicting pathways related to the nitrate respiration and nitrogen respiration were observably activated in the X50 group (*p* < 0.05). The X50 group improved the biological barrier, as manifested by increased *Firmicutes* and *Rhodobacter* (*p* < 0.05). In conclusion, dietary *C. vulgaris* powder could promote the growth performance of grass carp by restoring intestinal morphology, increasing digestive enzyme activities, improving antioxidant properties and immunity and optimizing the microflora structure. A *C. vulgaris* powder replacement of 50% soybean meal was recommended as feed for grass carp.

## 1. Introduction

Soybean meal is one of the major protein sources for herbivorous fishes [1] due to its high crude protein content and comparatively balanced amino acid composition [2]. However, some limitations to its utilization have been encountered, drawing attention to antinutritional factors, such as the urease, soybean lectin, trypsin inhibitor and so on, which negatively affect the intestinal health and reduce the utilization nutrients [3]. The huge gap between domestic soybean production and consumption had been filled by soybean import trade, which accounted for about 60% of the world’s annual imports [4]. Therefore, it is of great significance to develop an alternative protein source to reduce soybean meal use.

Single-cell microalgae are an emerging non-grain protein source with great development prospects. Some studies showed that microalgae promoted growth [5], improved immunity and antioxidant activity [6], restrained the growth of the pathogenic bacterium and ameliorated intestine health [7]. Among unicellular microalgae, *Chlorella vulgaris* (*C. vulgaris*) is well known for its high reproductive rate, short growth cycle, strong environmental tolerance, easy artificial cultivation and independence from geographical and climatic restrictions [8]. Additionally, *C. vulgaris* powder’s protein content reaches up to 50~60% and is rich in 18 kinds of amino acids. Especially, the threonine, glycine and proline contents of *C. vulgaris* are higher than those of soybean meal. Moreover, *C. vulgaris* powder contains a variety of beneficial substances, involving polysaccharides, pigments, vitamins, minerals and antioxidants [9]. Some related studies have verified that *C. vulgaris* promoted growth and relieved intestinal inflammation in African catfish (*Clarias gariepinus*) and Atlantic salmon (*Salmo salar* L.) [10,11]. It is worth mentioning that the polysaccharide, ferrum and aluminum accumulation in *C. vulgaris* powder inhibited the growth of fish [12] when they were fed diets with high levels of *C. vulgaris* powder inclusion. However, research about *C. vulgaris* powder inclusion in aquatic animals is still limited.

Grass carp (*Ctenopharyngodon idella*) is widely bred for its rapid growth, tender meat, rich nutrition and moderate price [13]. In practical production, the content of soybean meal as feed material in grass carp is 20~50%, accounting for about 21~53% of the cost. Therefore, this research aimed to explore the optimal proportion of *C. vulgaris* powder inclusion and its effect on the growth performance, intestinal structure, intestinal microbiota and immunity in grass carp. The results of this research will provide a theoretical foundation for the development and utilization of *C. vulgaris* powder as a novel non-grain protein source in aquatic animals.

## 2. Materials and Methods

### 2.1. Experimental Feed

Peruvian steamed fish meal, soybean meal and rapeseed meal were used as the main protein sources, and soybean oil was the lipid source. Based on isonitrogenous and isolipidic principles, the control group (SM) was set to contain 30.0% soybean meal, and then graded levels (25%, 50%, 75% and 100%) of soybean meal were replaced by *C. vulgaris* powder (designed as X25, X50, X75 and X100, respectively) (Table 1). The amino acids contents of *C. vulgaris* powder and soybean meal was in the Table 2. The ingredients were crushed, sieved (0.25 mm) and mixed with soybean oil. Water was added into the mixture. After being blended uniformly, the extruded feed (3.0 mm in particle size) was formed by a single-screw feed extruder system (DGP-100, TROT Co., Ltd., Hebei, China), and the feed was put in a shady place to dry naturally. The feed was stored at −20 °C.

### 2.2. Fish Experimental Design

Healthy grass carp (20.13 ± 0.09 g) were obtained and reared at Zhan Mao Lake, in Xidongting, Changde, Hunan province, China. The fish were allocated in two pond cages (3.0 m × 3.0 m × 3.0 m) for one week to adapt to laboratory conditions. A total of 750 grass carp were randomly assigned into five groups in triplicate. The SM, X25, X50, X75 or X100 diet was fed to different groups of fish for 8 weeks in the separate pond cages (2.0 m × 2.0 m × 2.0 m). The fish were artificially fed three times daily (8 a.m., 12 noon and 5 p.m.) at a 3~5% body weight. According to the feeding situation and the estimated weight of the grass carp, the feeding amount was adjusted once a week. The physicochemical index of pond water, such as the water temperature (28.4 ± 3.5 °C) (measured using a thermometer), pH value (7.6 ± 0.6), dissolved oxygen (7.2 ± 0.3 mg·L^−1^) and ammonia (≤0.2 mg·L^−1^) (using the LH-M900 portable colorimeter), were monitored. The water quality remained stable during the study period.

### 2.3. Sample Collection and Preservation

After the feeding trial, the fish were fasted for 24 h prior to collecting samples and then weighed to record the growth data. Three fish per net cage were chosen at random and then used to calculate the condition factors (CF), hepatosomatic index (HSI) and viscerosomatic index (VSI). The midgut samples were put in 10% neutral buffered formalin to make slices. The remaining midgut samples were used for the analysis of the mRNA expression levels and intestinal relate indices. The midgut contents from each fish were used for gut microflora analysis.

### 2.4. Sample Analyses

#### 2.4.1. Growth Index

The weight gain rate (WGR, %), feed conversion ratio (FCR), survival rate (SR, %), protein efficiency ratio (PER, %), condition factor (CF, g/cm^3^), hepatosomatic index (HSI, %) and viscerosomatic index (VSI, %) were calculated according to the following formulas:WGR = (W_F_ − W_0_)/W_0_ × 100
FCR = W_I_/(W_M_ − W_C_ + W_D_)
SR = N_M_/N_C_ × 100
PER = (W_M_ − W_C_)/(W_I_ × P_c_) × 100
CF = W_F_/(L_B_)^3^ × 100
HSI = W_G_/W_F_ × 100
VSI = W_N_/W_F_ × 100
where W_F_, W_O_, W_M_, W_C_, W_D_, W_I_, W_G_ and W_N_ were the last and original fish average weight, the last and original fish total weight, the weight of dead fish, the intake feed weight, the liver weight and the visceral weight; L_B_ was the fish body length; N_M_ and N_C_ were the last and original numbers of fish; T_A_ was the total feed consumed; P_c_ was the crude protein content of the feed.

#### 2.4.2. Intestinal Histopathological Examination

Intestinal samples were obtained from five groups of fish. First, the samples were fixed with 10% neutral buffered formalin. Second, the samples were dehydrated by a gradient with ethanol and xylene. Finally, the specimens were embedded in paraffin and stained with HE (hematoxylin-eosin staining) to make slides. The finished sections used the Case Viewer 2.0 analysis to measure the heights of villi, the muscle layer thickness and the number of goblet cells [14].

#### 2.4.3. Intestinal Immunity, Antioxidant Indices and mRNA Level Genes

The homogenizer (XHF-D, Xinzhi, China) was used to homogenize the intestine samples of fish with matching buffer. The intestine homogenate was centrifuged at 10,000× *g* rpm for 15 min at 4 °C. The supernatant of the samples was stored at −80 °C.

The kits of Nanjing Jiancheng Biological Engineering Research Institute (Nanjing, China) were used for determining the intestine superoxide dismutase (SOD) activity, catalase (CAT) activity, malondialdehyde (MDA) content, alkaline phosphatase (AKP) activity, acid phosphatase (ACP) activity, lysozyme (LZM) activity, amylase activity, lipases activity and trypsin activity.

According to the present growth and intestinal morphology results, the SM, X50 and X100 groups were selected to measure the intestinal related gene expression. Total RNA was extracted with TRIzol Reagent (Invitrogen, Carlsbad, CA, USA), according to the specification requirements. The RNA specimens of the intestine were analyzed by gel electrophoresis and put into the NanoDrop ND-2000 UV–Visible Spectrophotometer at 260 nm. The SMART cDNA Synthesis kit (Monad Biotech Co., Ltd., Beijing, China) was used to synthesize the cDNA with the total RNA. The cDNA was saved at −80 °C. Biosune Biotechnology, Inc. (Shanghai, China) provided the primers (Table 3). The reference gene was *β-actin.* Each assay had three replications. The E = 2^−ΔΔCT^ was the calculating formula.

#### 2.4.4. 16S rRNA Sequencing and Intestinal Microbiota Analysis

According to the present growth and intestinal morphology results, the SM, X50 and X100 groups were selected for 16S rRNA sequencing. The intestinal contents of three groups were extracted. The CTAB/SDS was used to extract the DNA of the specimens. The Illumina MiSeq platform used highthroughput sequencing and amplified the 16S rRNA V3-V4 region. The operational taxonomic units (OTUs) classified all the sequences. The 97% similarity level was selected by QIIME (version 2.0) after the FASTX-Toolkit (Hannon Lab, New York, NY, USA) removed low-quality scores (Q score, 20). UCLUST was used to classify each OTU. The above indices were calculated by QIIME and uploaded to NovoMagic (Beijing Novogene Technology Co., Ltd., Beijing, China) [15].

#### 2.4.5. Prediction of Intestinal Microbial Function 

The PICRUSt method (Phylogenetic Investigation of Communities by Reconstruction of Unobserved States) was used to predict the microbial function. The OTU abundance was the automatic normalization of the 16S rRNA gene copy number, which was from the Integrated Microbial Genomes. The predicted genes and related functions were matched to the KEGG (Kyoto Encyclopedia of Genes and Genomes) database.

### 2.5. Statistical Analysis

The experimental data were analyzed using SPSS version 27.0 (SPSS Inc., Munich, Germany), and the results were represented by the mean ± standard error of mean (SEM). One-way analysis of variance was used to analyze significant differences among mean values. Duncan’s multiple range test was used to determine the significant differences between experimental groups when *p* < 0.05.

## 3. Results

### 3.1. Feed Utilization, Indices, Growth and Biometric Parameters

At the end of the experimental trial, the X50 group showed higher FW, WGR and PER values than the SM group (*p* < 0.05). Compared to the control group, the PER was markedly reduced in the X100 group (*p* < 0.05). The fish from the experimental X50 group exhibited a lower FCR value when compared to that observed for the control group (*p* < 0.05). The CF decreased remarkably (*p* < 0.05) when soybean meal was replaced by *C. vulgaris* powder completely. There was no statistical difference in the SR, HSI and VSI among the five experimental groups (*p > 0.05*) (Table 4).

Compared to the control group, the crude protein content was considerably increased when soybean meal was replaced by *C. vulgaris* powder (*p* < 0.05). The moisture content increased substantially in the X50 and X100 groups (*p* < 0.05). There was no statistical difference in the crude lipid and ash contents among the five experimental groups (*p > 0.05*) (Table 5).

### 3.2. Intestinal Morphology and Digestive Enzyme Activities

The villus height of grass carp increased as the *C. vulgaris* powder substitution levels increased, and more than 25% of the soybean meal replaced by *C. vulgaris* powder was higher than that of the SM group (*p* < 0.05). The number of goblet cells revealed a significant increase in the X50 group in comparison with the SM group (*p* < 0.05). The muscle thickness was independent of the dietary *C. vulgaris* powder (*p* > 0.05). The middle intestinal villi in the SM and X25 groups were slightly damaged. In contrast, a markedly reduced injury of the middle intestinal villi was observed in the X50 group. However, the intestinal villi in the X75 and X100 groups were structurally disordered and showed intestinal villus adhesion, goblet cells hyperplasia and capillary congestion (Table 6, Figure 1).

The amylase and trypsin activities of grass carp in the X50 group were substantially (*p* < 0.05) increased compared with those of the SM group. Compared to that of the control group, the trypsin activity in the X100 was notably reduced. There was no statistical difference in the lipases of grass carp among the five experimental groups (*p* > 0.05) (Table 7).

It was observed that the *zo-1*, *zo-2* and *occludin* of the mRNA expression levels were significantly upregulated in the X50 group compared with those in the SM group (*p* < 0.05). Meanwhile, the mRNA expression level of the *claudin12* showed no significant differences in the X50 group compared with that of the SM group (*p* > 0.05). The *claudin12* was markedly downregulated, and there was no significant differences in the other mRNA expression levels in the X100 group compared with those in the SM group (*p* > 0.05) (Figure 2).

### 3.3. Intestine Antioxidant Indices

The highest SOD value was observed in the X50 group when compared with the control group (*p* < 0.05). The CAT activity decreased in the group of fish fed diets with more than 50% soybean replacement by *C. vulgaris* powder when compared with the SM group (*p* < 0.05). The MDA content was observably (*p* < 0.05) decreased in X25, X50 and X75 groups (*p* < 0.05) compared with that of the SM group (Table 8).

It was observed that the *mnsod* and *nrf2* of the mRNA expression levels were markedly upregulated, while the *keap1* was remarkably downregulated in the X50 group compared with the SM group (*p* < 0.05). Meanwhile, the mRNA expression levels of the *cat* and *cuznsod* showed no significant differences in the X50 group compared with those of the control group (*p* > 0.05). There were no significant differences in SOD and CAT between the SM group and X100 group (Figure 3).

### 3.4. Intestine Immune Responses and Inflammation

The AKP, ACP and LZM activities were higher in the X50 group compared to those in the SM group (*p* < 0.05). The AKP activity was considerably (*p* < 0.05) decreased in the X50, X75 and X100 groups (*p* < 0.05) compared with that of the SM group (Table 9).

It was observed that the *tlr-4*, *tlr-8* and *il-6* were substantially downregulated in the X50 group compared with those in the SM group (*p* < 0.05). Meanwhile, the mRNA expression levels of the *tlr-7* and *il-1β* showed no significant differences in the X50 group compared with the SM group (*p* > 0.05). The *tlr-7* was appreciably upregulated, and there were no significant differences in other mRNA expression levels in the X100 group compared with those of the SM group (*p* > 0.05) (Figure 4).

### 3.5. Intestinal Microbiota

#### 3.5.1. Comparison of Abundance and Diversity

The predominant intestinal microbiota were identified, including *Firmicutes*, *Bacteroidetes*, *Proteobacteria* and *Fusobacteria*. *Bacteroidetes* showed the highest relative abundance at 40.45% in the X100 group, while *Firmicutes* showed the highest relative abundance at 57.26% in the X50 group (*p* < 0.05) (Figure 5).

There was no statistical difference in the intestinal microbial population in the Chao1, Shannon and Simpson of grass carp among the groups (*p* > 0.05) (Figure 6).

#### 3.5.2. Comparison of the Microbial Community Structure

Unweighted beta-diversity analysis showed a clear shift in the cluster of microbiota among the three experimental groups through the principal coordinates analysis (PCoA). In terms of cluster analysis, the *C. vulgaris* powder replacement of 50% soybean meal showed the largest distance from the control group (*p* < 0.05) (Figure 7).

For LEfSe, a differential abundance of bacterial taxa among different *C. vulgaris* powder treatments was spotted. As seen in Figure 8A, the phylogenetic composition of OTUs was noticeably different among *C. vulgaris* powder treatment samples. As shown in Figure 8B, a total of 25 bacterial biomarkers were differentially abundant among the three experimental groups. In comparison, *Desulfobacterota* (including the *Desulfobaccaceae*, *Desulfobaccales* and *Desulfobaccia*) were the abundant taxa in the SM group, while that of the X100 group was *Fusobacteriota* (including the *Fusobacteriales* and *Fusobacteriales*) (*p* < 0.05) (Figure 8A). The SM group had three greater differentially abundant genera, including *Alsobacter*, *Desulfobacca* and UTCFX1(*p* < 0.05). In addition, only *Rhodobacter* in the X50 group and *Akkermansia* in the X100 group were the most differentially abundant genera (*p* < 0.05) (Figure 8B).

#### 3.5.3. Functional Predictions of Intestinal Microbiota

A deeper analysis of the KEGG among the three experimental groups revealed that the nitrate respiration and nitrogen respiration were appreciably enhanced in the X50 group (*p* < 0.05) (Figure 9A). The fermentation and chemoheterotrophy were observably enriched in the X100 group compared with those of the SM and X50 groups (*p* < 0.05) (Figure 9B,C). 

## 4. Discussion

In the present study, the growth performance was observably improved in the X50 group of grass carp, which was similar to the results conducted on Nile tilapia (*Oreochromis niloticus*) [16]. This was likely because the higher content of threonine, glycine and proline in *C. vulgaris* powder relative to soybean meal promoted grass carp growth, which was consistent with the results in shrimp (*Penaeus vannamei*) and large yellow croaker (*Larimichthys crocea*) [17,18]. Moreover, *C. vulgaris* was rich in unsaturated fatty acids and the *Chlorella* growth factor (CGF), which was beneficial to fish growth and protein efficiency [19]. Furthermore, the antinutritional factors decreased with the decrease in the soybean meal content, which may be one of the reasons for the best growth of grass carp in the X50 group [20]. However, the total substitution of *C. vulgaris* powder with soybean meal reduced the growth performance of grass carp. This was in agreement with the result of juvenile yellow perch (*Perca flavescens*) [21] and blunt snout bream (*Megalobrama amblycephala*) [22]. This might be caused by a negative feedback system for a high concentration of active polysaccharides from high levels of *C. vulgaris* powder in the body, which made the growth of fish back to a normal level or even below a normal level [23]. In addition, ferrum and aluminum accumulation with high levels *C. vulgaris* powder affected the feed utilization in the intestine by preventing the absorption of potassium, calcium, magnesium and other elements of grass carp, which was unfavorable for growing [24]. An increased proportion of *C. vulgaris* led to an amino acid imbalance, which was also one of the reasons [25].

Additionally, the intestine plays a crucial role in the digestion and absorption of nutrients directly defining the growth of grass carp [26]. The intestinal villi are formed by the surface epithelium and the lamina propria below it, protruding into the lumen. Higher villi indicated faster tissue turnover for permitting the renewal of the intestinal epithelium, which increased the contact area to improve nutrient absorption [27]. In this experiment, the villus height increased when the 50% soybean meal was replaced by *C. vulgaris* powder. It was the same with the study of *Spirulina* and *Chlorella* by-products in broiler chickens [28,29]. However, high levels of *C. vulgaris* powder replacement of soybean meal showed intestinal villus adhesion, goblet cells hyperplasia and capillary congestion. It may be related to the content of metal elements in *C. vulgaris* [30]. Digestive enzymes facilitate the absorption of nutrients by fish [31]. In this study, digestive enzyme activities increased when *C. vulgaris* powder replaced the 50% soybean meal. This was consistent with the results of giant freshwater prawn (*Macrobrachium rosenbergii*) and grey mullet (*Mugil cephalus*) [32,33]. Supposedly, the improved growth by the *C. vulgaris* powder replacement of soybean meal might be related to the increase in intestinal villi and digestive enzyme activities.

The tight junction is a complex structure formed by the interaction of multiple proteins for closing the spaces between intestinal epithelial cells [34]. Among tight junctions between cells, *zo-1*, *zo-2* and *occludin* were important for restoring the intestinal mechanical barrier [35]. In this study, the mRNA expression levels of the *zo-1*, *zo-2* and *occludin* genes were upregulated when the 50% soybean meal was replaced by a *C. vulgaris* powder diet, demonstrating the improved integrity of the mucosal barrier structure. This result was similar to the study in mice in which that microalgae improved the intestinal structure through improving the tight junction proteins expression [36]. Goblet cells secrete mucin and trefoil peptides to protect the mucosal structure, which covers on the surface of the intestinal epithelium to strengthen intestinal mechanical barrier function [37]. In the study, the variation in the number of goblet cells was consistent with the expression of the tight junction protein. It was speculated that the increase in goblet cells may be related to the improvement of the mucosal structural integrity induced by *C. vulgaris*.

The antioxidative defense system can scavenge oxygen free radicals constantly produced by the body in the process of metabolism in order to keep the homeostatic regulation [38]. The X50 group’s antioxidant capacity was improved. Likewise, *C. vulgaris* and *Chlorella pyrenoidosa* polysaccharides contributed to the antioxidative capacity [39,40]. The polyphenols [41], flavonoids [42] and phytopigments [43] in *C. vulgaris* were able to chelate redox-active metals and accept electrons from reactive oxygen species. This may be a reason for the enhanced antioxidant activity in the *C. vulgaris*, whereas the *C. vulgaris* powder completely replacing the soybean meal showed an opposite trend. Similar to this experiment, antioxidant enzyme activities reduced with higher dietary levels of *Nannochloropsis* [44,45]. The *Nrf2-keap1* signaling pathway maintains the oxidation-reduction reaction balance and metabolism of cells [46]. In the present study, *keap1* was downregulated and *nrf2* was upregulated in the X50 group, which was in line with the changes in studies in rats [47]. The results revealed that the *C. vulgaris* powder of proper substitution levels could promote the intestinal antioxidant capacity through the activation of the *nrf2-keap1* signaling pathway.

AKP and ACP are two important hydrolytic enzymes which are involved in the metastasis and metabolism of the phosphate group. LZM dissolves cell walls to engulf bacteria and mediates protection against microbial invasion. The three enzyme activities are common indexes for assessing the nonspecific immune function of the body [48]. A variety of evidence has demonstrated dietary *C. vulgaris* to be a potent immunostimulant on juvenile rainbow trout (*Oncorhynchus mykiss*) [49], Nile tilapia [50] and gibel carp (*Carassius auratus gibelio*) [51]. In this experiment, the ACP and LZM activities in the X50 group increased. This may be ascribed to the involvement of polysaccharides and the *Chlorella* growth factor (CGF) in regulating the non-specific immune response of grass carp [52].

The inflammatory response regulates the immune system by enhancing or inhibiting the expression levels of cytokines [53]. The regulation of microalgae regarding the inflammatory response has been reported. *Chlorella pyrenoidosa* decreased the levels of the inflammatory factor *il-6* in the serum of rats with a high-fat diet [54]. The microalgae aqueous extracts inhibited inflammatory effects in *il-1β*, stimulating Caco-2 cells [55]. In this study, the mRNA expression levels of *il-6* and *il-1β* were downregulated after the 50% soybean meal was replaced by a *C. vulgaris* powder diet. This was similar to the results in the glycolytic enzyme extract of microalgae residues [56]. The findings showed that moderate *C. vulgaris* powder replacing soybean meal could relieve the intestinal inflammation of grass carp. Toll-like receptors (*tlrs*) played a pivotal role in regulating the secretion of inflammatory cytokines and the activation of the inflammatory response [57]. In the present study, the mRNA expression levels of *tlr-4*, *tlr-7* and *tlr-8* were downregulated after the 50% soybean meal was replaced by a *C. vulgaris* powder diet. This was consistent with the gibel carp fed with dietary *Scenedesmus ovalternus* [58]. The proinflammatory cytokines expression was closely correlated with the *tlrs*, revealing that the anti-inflammation of *C. vulgaris* may be partially via *tlrs* signaling.

Intestinal microbial populations colonize the intestinal tract, forming a biological barrier which influences various physiological processes [59]. The *Firmicutes* are mostly beneficial bacteria, and the *Bacteroidetes* can produce enterotoxins. Compared with the group X50, the abundance of the *Firmicutes* decreased and the *Bacteroidetes* increased in group X100. This was a reason why the growth performance of the X50 group was better than that of the X100 group [60]. It was shown that the intestinal microbial intricacy and abundance were affected by the *C. vulgaris* powder replacement of the 50% soybean meal. The same observation could be seen in the study of *Chlorella pyrenoidosa* regarding the human colon [61]. LEfse showed that the *Desulfobaccerota* was remarkedly reduced in the X100 group. The *Desulfobacterium* played an essential role in the butyrate metabolism [62]. The butyrate is a key compound for both microbial and body epithelial cell growth, which improves the growth performance of Chinese striped-neck turtles (*Mauremys sinensis*) with dietary supplementation [63]. Therefore, it was inferred that the *Desulfobacterium* producing butyrate metabolites may affect the growth performance of grass carp with an excessive *C. vulgaris* powder replacement of soybean meal. Moreover, the *Fusobacteriota* was enriched in the X100 group of intestinal microbials. *Fusobacteriota* was connected with diarrhea and enteritis in newborn piglets [64,65]. These results indicate that the *C. vulgaris* complete replacement of soybean meal could also influence the intestinal homeostasis caused by enriching conditioned pathogens. Meanwhile, at the genus level, the results displayed that *Rhodobacter* increased remarkably in the X50 group of intestinal microbials. *Rhodobacter* is a kind of probiotic bacteria which contributes to promoting the growth of seawater red tilapia [66]. The *Rhodobacter sphaeroides* protein maintained the intestinal health [67]. Therefore, it was inferred that appropriate *C. vulgaris* powder strengthened the intestinal biologic barrier to limit the access of pathogenic bacteria and maintain intestinal health, contributing to the growth performance of grass carp. 

## 5. Conclusions

In summary, the optimum *C. vulgaris* powder replacement of soybean meal improved the growth performance of grass carp by restoring the intestinal morphology, increasing digestive enzyme activities, improving antioxidant properties and immunity and optimizing the microflora structure. However, the positive effects were apparently weakened when the soybean meal was replaced with high levels of *C. vulgaris* powder. Therefore, *C. vulgaris* powder replacement of 50% soybean meal was recommended as feed for grass carp. This study provided a reference for *C. vulgaris* as a novel non-grain single-cell protein source for replacing soybean meal in aquatic animals.

## Figures and Tables

**Figure 1 animals-13-02274-f001:**
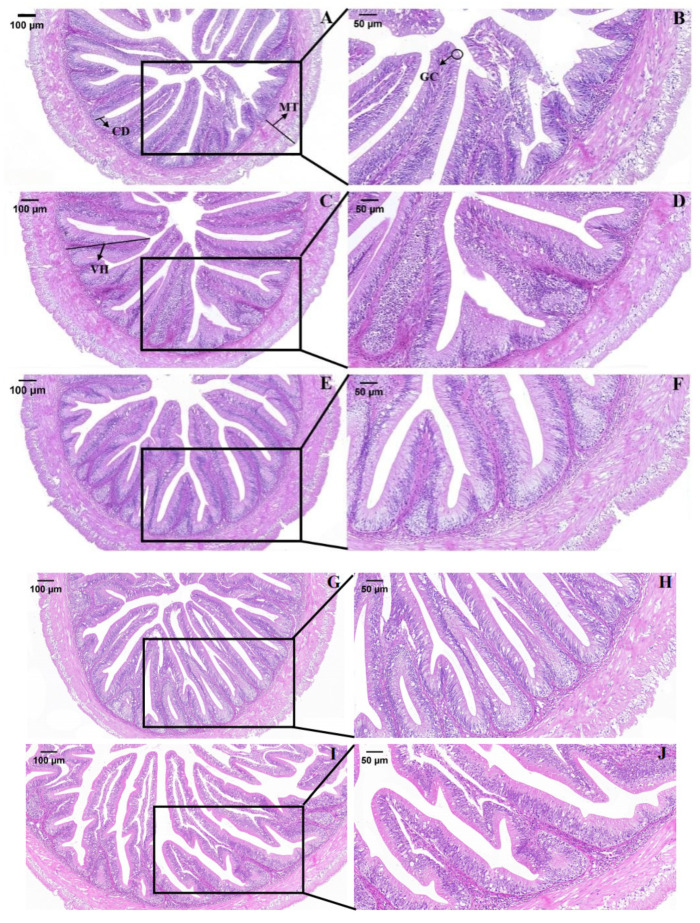
Mucosa morphology of middle intestinal (**A**,**C**,**E**,**G**,**I**) SM, X25, X50, X75 and X100 groups (Magnification ×100), respectively. (**B**,**D**,**F**,**H**,**J**) show the partial enlargement (Magnification ×200). VH: villus height, MT: muscle thickness, GC: goblet cell.

**Figure 2 animals-13-02274-f002:**
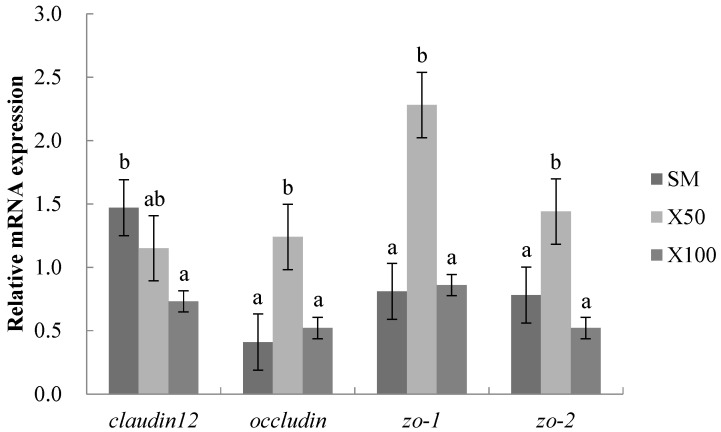
Effects of the *C. vulgaris* powder replacement of soybean meal on the intestinal relative mRNA expression in grass carp after 8 weeks (Mean ± SEM, *n* = 3). The statistical differences were represented by different superscripts (*p* < 0.05) (*p* = 0.082, 0.044, 0.001 and 0.014, respectively).

**Figure 3 animals-13-02274-f003:**
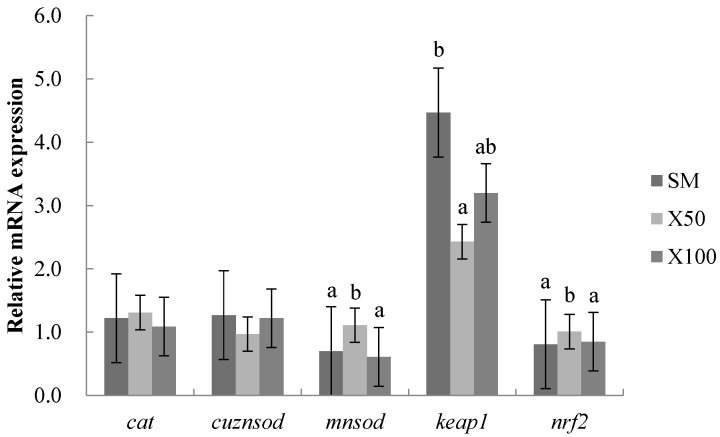
Effects of *C. vulgaris* powder replacement of soybean meal on the intestinal relative mRNA expression in grass carp after 8 weeks (Mean ± SEM, *n* = 3). The statistical differences are represented by different superscripts (*p* < 0.05) (*p* = 0.619, 0.236, 0.02, 0.064 and 0.01, respectively).

**Figure 4 animals-13-02274-f004:**
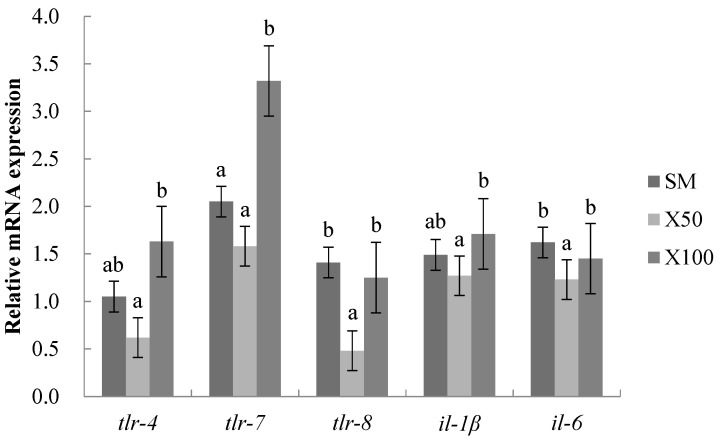
Effects of *C. vulgaris* powder replacement of soybean meal on intestinal relative mRNA expression in grass carp after 8 weeks (Mean ± SEM, *n* = 3). The statistical differences are represented by different superscripts (*p* < 0.05) (*p* = 0.023, 0.001, 0.027, 0.039 and 0.007, respectively).

**Figure 5 animals-13-02274-f005:**
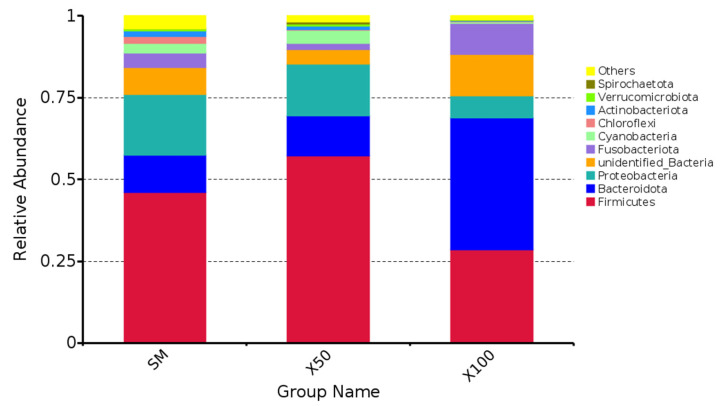
Community taxonomy composition and abundance map of grass carp with different levels of *C. vulgaris* powder replacing soybean meal at the phylum level (*n* = 3).

**Figure 6 animals-13-02274-f006:**
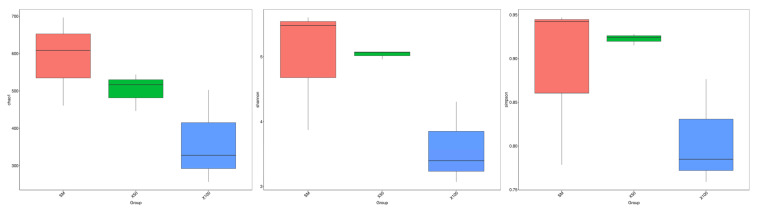
The α-diversity indices of the bacterial community of grass carp with different levels of *C. vulgaris* powder replacing soybean meal (*n* = 3), and box plots showing the Chao1, Shannon and Simpson indices in the intestine microbiota.

**Figure 7 animals-13-02274-f007:**
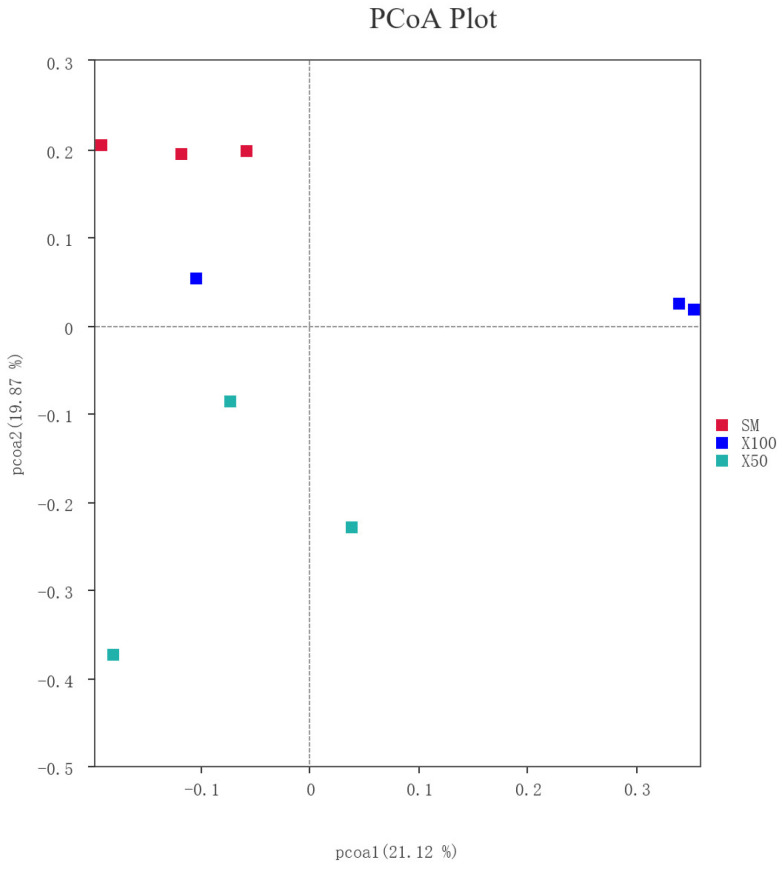
The principal coordinates analysis (PCoA) of bacterial OTUs showed an intestinal microbial cluster in the SM, X50 and X100 groups.

**Figure 8 animals-13-02274-f008:**
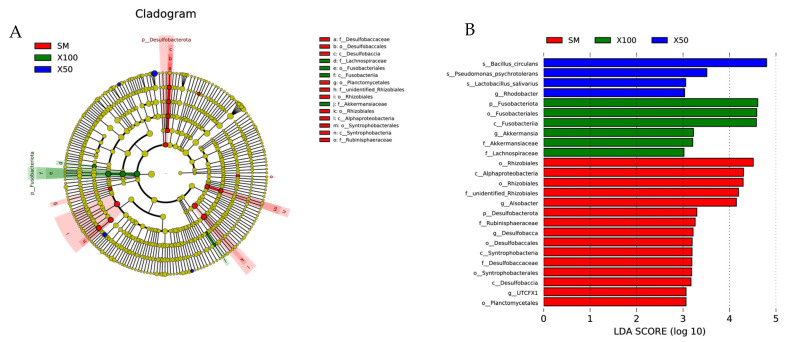
LEfSe manifested as a feature characterized by the intestinal microbial population, it was performed to identify differential taxa among gut microflora in grass carp with different levels of *C. vulgaris* powder replacement of soybean meal (*n* = 3). The dots in the center present OTUs at phylum levels, whereas the outer circle of dots present the OTUs at genus levels. Different colors of the dots and sectors indicate the compartment in which the respective OTUs are most abundant. The upper left corner of the legend is the color explanation. The yellow color indicates that the OTUs revealed a similar abundance in all compartments (**A**). The most differentially abundant intestinal microbial was characterized in the SM, X50 and X100 groups of grass carp (**B**).

**Figure 9 animals-13-02274-f009:**
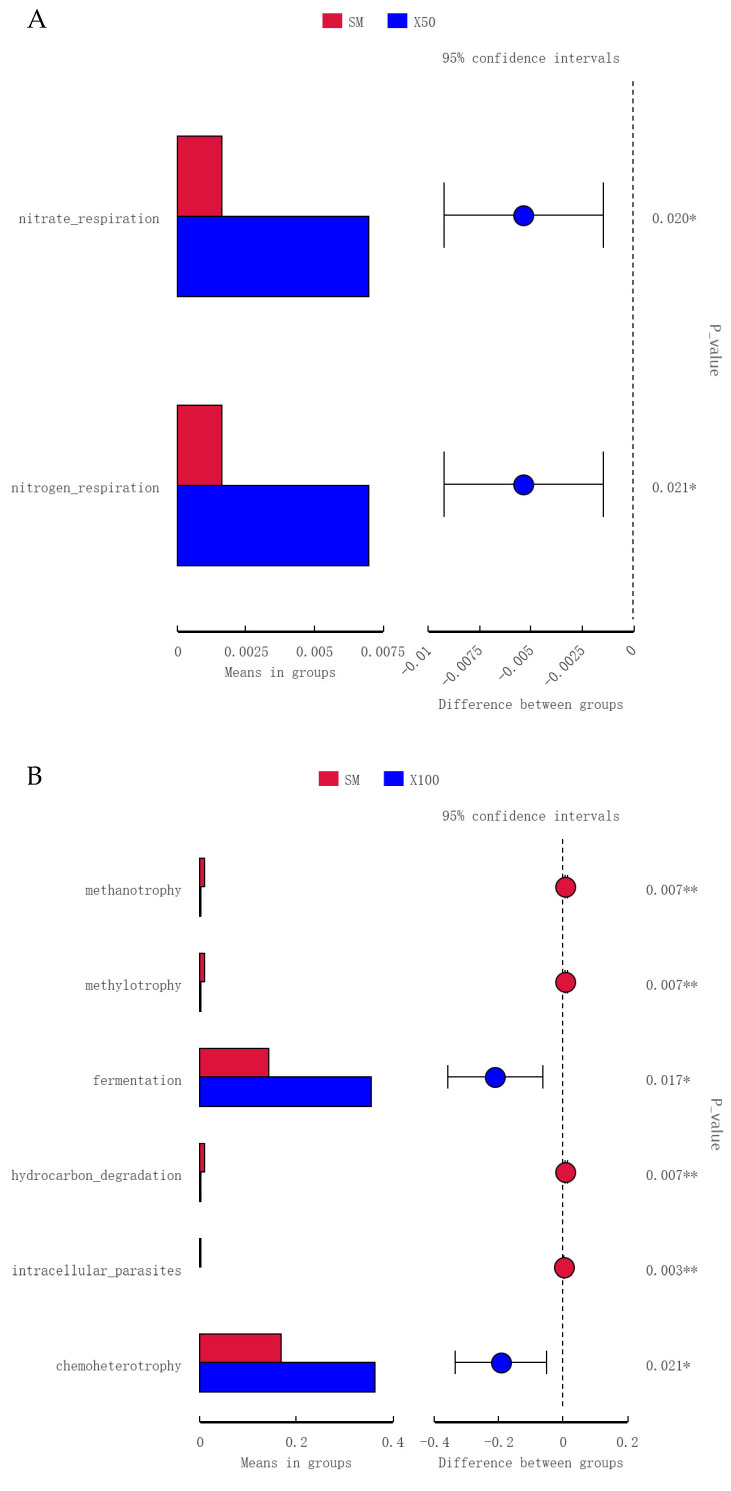
Extended error bar plot revealing the significantly different KEGG metabolic pathways among the SM, X50 and X100 groups. (**A**) SM vs. X50, (**B**) SM vs. X100 and (**C**) X50 vs. X100 intestine of grass carp. ((**A**): Red was the SM group, blue was the X50 group; (**B**): Red was the SM group, blue was the X100 group; (**C**): Red was the X50 group, blue was the X100 group. They were the error bars in the circle). * *p* < 0.05, ** *p* < 0.01.

**Table 1 animals-13-02274-t001:** Formulation and proximate composition of nutrition (g/kg).

Ingredients	SM	X25	X50	X75	X100
Fish meal	40.0	40.0	40.0	40.0	40.0
Distiller dried grains with solubles	90.0	90.0	90.0	90.0	90.0
Soybean meal ^1^	300.0	225.0	150.0	75.0	0.0
*C. vulgaris* powder ^2^	0.0	60.1	120.1	180.2	240.3
Rapeseed meal	200.0	200.0	200.0	200.0	200.0
Rice bran	100.0	100.0	100.0	100.0	100.0
Wheat flour	220.0	220.0	220.0	220.0	220.0
Soybean oil	21.4	16.0	10.7	5.3	0.0
Bentonite	1.2	21.5	41.8	62.1	82.3
Choline chloride	2.0	2.0	2.0	2.0	2.0
Ca(H_2_PO_4_)_2_	15.0	15.0	15.0	15.0	15.0
Premix ^3^	10.0	10.0	10.0	10.0	10.0
Antioxidant	0.1	0.1	0.1	0.1	0.1
Anti-mildew agent	0.3	0.3	0.3	0.3	0.3
Total	1000.0	1000.0	1000.0	1000.0	1000.0
Proximate analysis					
Crude protein	304.6	306.9	305.9	306.7	307.8
Crude fat	55.4	56.6	56.8	57.2	57.8
Crude ash	102.9	99.4	101.3	100.5	102.8

^1^ Soybean meal was USA soybean meal, which was provided by Tongwei Co., Ltd. (Chengdu, China) (Crude protein 46.05%, Crude fat 1.03%). ^2^
*C. vulgaris* powder was provided by Wuhan Demeter Biotechnology Co., Ltd. (Wuhan, China) (Crude protein 57.50%, Crude fat 10.02%). ^3^ Premix was provided by MGOTer Bio-Tech Co., Ltd. (Qingdao, China). Premix composition (mg/kg diet): vitamin A, 120,000 IU; vitamin D_3_, 40,000 IU; VC phosphatase, 6850 mg; iron, 4800 mg; magnesium, 4000 mg; acid, 3200 mg; zinc, 2000 mg; nicotinic acid, 1000 mg; manganese, 800 mg; calcium pantothenate, 720 mg; vitamin E, 480 mg; vitamin B_2_, 280 mg; 240 mg; vitamin B_1_, 200 mg; vitamin K_3_, 200 mg; copper, 160 mg; folic acid, 60 mg; iodine, 40 mg; cobalt, 12 mg; selenium, 4 mg; biotin, 1.2 mg; vitamin B_12_, 0.6 mg.

**Table 2 animals-13-02274-t002:** The amino acids contents of *C. vulgaris* powder and soybean meal (g/kg).

Amino Acids	*C. vulgaris*	Soybean Meal
Ala	38.0	20.4
Arg ^Δ^	27.1	34.6
Asp	42.2	54.0
Cys	2.4	6.8
Glu	69.6	83.0
Gly	24.2	19.7
His ^Δ^	10.4	12.5
Ile ^Δ^	15.0	20.4
Leu ^Δ^	36.6	34.7
Lys ^Δ^	27.6	24.7
Met ^Δ^	6.1	6.1
Phe ^Δ^	22.1	23.4
Pro	33.1	23.5
Ser	20.6	24.2
Thr ^Δ^	23.3	18.3
Trp	/	6.5
Tyr	14.6	/
Val ^Δ^	26.0	22.8

^Δ^ For essential amino acids. The amino acids were detected by the automatic amino acid analyzer (Agilent-1100, Wuhan Demot Biotechnology Co., Ltd., Santa Clara, CA, USA).

**Table 3 animals-13-02274-t003:** Primer sequences of qRT-PCR.

Gene	Forward (5′-3′)	Reverse (5′-3′)	Accession No.
*β-actin*	GATGATGAAATTGCCGCACTG	ACCGACCATGACGCCCTGATGT	M25013
*cat*	GAAGTTCTACACCGATGAGG	CCAGAAATCCCAAACCAT	FJ560431
*cuznsod*	CGCACTTCAACCCTTACA	ACTTTCCTCATTGCCTCC	GU901214
*mnsod*	ACGACCCAAGTCTCCCTA	ACCCTGTGGTTCTCCTCC	GU218534
*nrf2*	CTGGACGAGGAGACTGGA	ATCTGTGGTAGGTGGAAC	KF733814
*keap1*	TTCCACGCCCTCCTCAA	TGTACCCTCCCGCTATG	KF811013
*il-1* *β*	AGAGTTTGGTGAAGAAGAGG	TTATTGTGGTTACGCTGGA	JQ692172
*il-6*	CAGCAGAATGGGGGAGTTATC	CTCGCAGAGTCTTGACATCCTT	KC535507.1
*tlr* *-* *4*	TTCCACCTATTCATCTTTGC	ACTTTACGGCTGCCCATT	EU699768.1
*tlr* *-7*	GAGCATACAGTTGAGTAAACGCAC	TCTCCAAGAATATCAGGACGATAA	JN867639.1
*tlr* *-8*	TCACATCGCTTCCAGGTCTC	ACGGTGAAATAATGGGGGTT	HQ638214.1
*occludin*	TATCTGTATCACTACTGCGTCG	CATTCACCCAATCCTCCA	KF193855.1
*claudin12*	CCCTGAAGTGCCCACAA	GCGTATGTCACGGGAGAA	KF998571
*zo-1*	CGGTGTCTTCGTAGTCGG	CAGTTGGTTTGGGTTTCAG	KF193852.1
*zo-2*	TACAGCGGGACTCTAAAATGG	TCACACGGTCGTTCTCAAAG	KM112095

**Table 4 animals-13-02274-t004:** Effects of soybean meal replacement by *C. vulgaris* powder on the growth indices of grass carp after eight weeks (Mean ± SEM, *n* = 3).

Items	SM	X25	X50	X75	X100
IW ^1^	20.28 ± 0.18	20.04 ± 0.13	20.11 ± 0.13	20.16 ± 0.10	20.08 ± 0.18
FW ^2^	60.11 ± 1.13 ^a^	57.87 ± 3.96 ^a^	67.25 ± 0.49 ^b^	61.57 ± 0.48 ^ab^	59.43 ± 1.10 ^a^
WGR ^3^	200.56 ± 1.15 ^a^	189.34 ± 19.78 ^a^	232.92 ± 2.67 ^b^	207.83 ± 2.38 ^ab^	197.16 ± 5.49 ^a^
FCR ^4^	1.71 ± 0.04 ^bc^	1.64 ± 0.05 ^b^	1.45 ± 0.03 ^a^	1.58 ± 0.03 ^b^	1.79 ± 0.04 ^c^
SR ^5^	87.33 ± 0.67	87.33 ± 1.76	87.33 ± 2.91	86.00 ± 2.00	89.33 ± 2.40
PER ^6^	168.89 ± 4.11 ^b^	159.56 ± 7.03 ^b^	208.58 ± 1.98 ^c^	174.27 ± 3.41 ^b^	142.23 ± 5.09 ^a^
CF ^7^	2.18 ± 0.05 ^b^	1.99 ± 0.04 ^ab^	2.06 ± 0.05 ^ab^	2.06 ± 0.04 ^ab^	1.92 ± 0.12 ^a^
HIS ^8^	3.45 ± 0.12	3.69 ± 0.23	3.59 ± 0.34	3.45 ± 0.26	3.84 ± 0.48
VSI ^9^	20.63 ± 0.62	19.17 ± 0.41	21.68 ± 0.68	18.32 ± 3.71	19.27 ± 1.74

The statistical differences are represented by different superscripts at *p* < 0.05 (Duncan test). ^1^ IW: Initial body weight (g); ^2^ FW: Final body weight (g); ^3^ WGR: Weight gain rate (%); ^4^ FCR: Feed conversion rate; ^5^ SR: Survival rate (%); ^6^ PER: Protein efficiency ratio (%); ^7^ CF: Condition factor (%); ^8^ HSI: Hepatopancreas index (%); ^9^ VSI: Viscerosomatic index (%).

**Table 5 animals-13-02274-t005:** Effects of soybean meal replacement by *C. vulgaris* powder on the body composition of grass carp after eight weeks (Mean ± SEM, *n* = 3) (%).

Items	SM	X25	X50	X75	X100
Crude protein	17.63 ± 0.41 ^a^	19.34 ± 0.30 ^b^	20.62 ± 0.31 ^c^	19.61 ± 0.54 ^bc^	20.76 ± 0.31 ^c^
Crude lipid	8.27 ± 0.50	7.38 ± 0.51	7.23 ± 0.15	7.13 ± 0.55	7.33 ± 0.60
Moisture	68.61 ± 0.16 ^a^	68.96 ± 0.58 ^a^	70.56 ± 0.30 ^b^	69.75 ± 0.66 ^ab^	70.48 ± 0.28 ^b^
Ash	3.95 ± 0.26	3.09 ± 0.80	1.65 ± 0.58	3.49 ± 1.11	1.64 ± 0.46

The statistical differences are represented by different superscripts at *p* < 0.05 (Duncan test).

**Table 6 animals-13-02274-t006:** Effects of soybean meal replacement by *C. vulgaris* powder on the intestinal morphology of grass carp after eight weeks (Mean ± SEM, *n* = 3).

Items	SM	X25	X50	X75	X100
Villus height (μm)	510.03 ± 24.41 ^a^	586.43 ± 28.20 ^ab^	601.27 ± 23.30 ^b^	605.60 ± 30.50 ^b^	604.90 ± 18.36 ^b^
Muscle thickness (μm)	206.60 ± 13.61	207.81 ± 12.11	199.87 ± 5.34	198.34 ± 13.08	195.06 ± 7.72
Goblet cell (A/root)	43.00 ± 3.21 ^b^	51.00 ± 3.21 ^bc^	53.33 ± 4.81 ^c^	31.67 ± 0.88 ^a^	31.67 ± 1.45 ^a^

The statistical differences are represented by different superscripts at *p* < 0.05 (Duncan test).

**Table 7 animals-13-02274-t007:** Effects of soybean meal replacement by *C. vulgaris* powder on the digestive enzyme activities of grass carp in the intestine after eight weeks (Mean ± SEM, *n* = 3).

Items	SM	X25	X50	X75	X100
Amylase (U/mg prot)	8.50 ± 0.30 ^a^	8.16 ± 0.67 ^a^	12.00 ± 0.79 ^b^	9.31 ± 0.42 ^a^	8.75 ± 1.42 ^a^
Lipases (U/g prot)	6.96 ± 0.40	5.89 ± 0.60	6.03 ± 0.85	5.10 ± 0.63	5.25 ± 0.60
Trypsin (U/mg prot)	573.80 ± 20.07 ^b^	543.33 ± 17.92 ^ab^	656.80 ± 13.66 ^c^	522.86 ± 15.79 ^ab^	490.80 ± 14.17 ^a^

The statistical differences are represented by different superscripts at *p* < 0.05 (Duncan test).

**Table 8 animals-13-02274-t008:** Effects of soybean meal replacement by *C. vulgaris* powder on the antioxidant index of grass carp in the intestine after eight weeks (Mean ± SEM, *n* = 3).

Items	SM	X25	X50	X75	X100
SOD ^1^	87.56 ± 4.52 ^a^	104.61 ± 5.22 ^ab^	112.79 ± 4.31 ^b^	96.05 ± 6.79 ^ab^	93.64 ± 6.56 ^a^
CAT ^2^	21.46 ± 14.32 ^b^	22.80 ± 20.62 ^b^	24.80 ± 21.86 ^b^	15.87 ± 13.60 ^a^	16.83 ± 11.50 ^a^
MDA ^3^	5.32 ± 0.08 ^c^	3.21 ± 0.48 ^b^	2.36 ± 0.05 ^a^	3.79 ± 0.09 ^b^	5.00 ± 0.08 ^c^

The statistical differences are represented by different superscripts at *p* < 0.05 (Duncan test). ^1^ SOD: Superoxide dismutase (U/mg); ^2^ CAT: Catalase (U/mg); ^3^ MDA: Malondialdehyde (nmol/mg).

**Table 9 animals-13-02274-t009:** Effects of soybean meal replacement by *C. vulgaris* powder on the immune enzyme activities of grass carp in the intestine after eight weeks (Mean ± SEM, *n* = 3).

Items	SM	X25	X50	X75	X100
AKP ^1^	259.10 ± 13.57 ^c^	230.77 ± 2.85 ^bc^	206.83 ± 17.91 ^b^	112.45 ± 2.74 ^a^	112.67 ± 4.91 ^a^
ACP ^2^	224.92 ± 6.10 ^ab^	253.60 ± 19.89 ^b^	302.61 ± 4.57 ^c^	193.77 ± 12.77 ^a^	195.80 ± 14.52 ^a^
LZM ^3^	31.82 ± 2.62 ^ab^	43.94 ± 4.01 ^c^	53.41 ± 0.66 ^d^	34.09 ± 0.58 ^b^	26.14 ± 0.66 ^a^

The statistical differences are represented by different superscripts at *p* < 0.05 (Duncan test). ^1^ AKP: Alkaline phosphatase (King’s unit/g); ^2^ ACP: Acid phosphatase (King’s unit/g; ^3^ LZM: Lysozyme (U/mg).

## Data Availability

The data presented in this study are available on request from the corresponding author.

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
