# Peer review of "Effects of Replacing Soybean Meal Protein with Chlorella vulgaris Powder on the Growth and Intestinal Health of Grass Carp (Ctenopharyngodon idella)"

_animals, 2023, doi:10.3390/ani13142274_

Round 1

Reviewer 1 Report

The purpose of the experiment was to explore the influence of the substitution of soybean meal with Chlorella vulgaris on grass carp (Ctenopharyngodon idella) in terms of growth performance, intestinal integrity and microbial community. The results are practical valuable for the application of Chlorella vulgaris in feed of grass carp. However, some sections of this manuscript need to be improved. The following are the major comments for references.

Comment 1: Line 117-118. Is the temperature of the pond measured by how many surface water or centimeters of underwater? Whether the dissolved oxygen (7.2 ± 0.3 mg·L1) conforms to the actual situation.

Comment 2: Line 143, “intake feed weights” should be replaced by “intake feed weight”.

Comment 3: Line 150---add “specimens” instead of “samples”.

Comment 4: Line 280-301. What are the indicators related to intestinal inflammation? Does gene expression alone mean grass carp have an inflammatory response?

Comment 5: Line 303, “richness” should be replaced by “abundance”, both of which the concepts are different.

Comment 6: Line 423, “engulf bacteriais” should be replaced by “engulf bacteria”. Some words are not standard, please check.

Comment 7: The manuscript needs thorough English proofreading for grammatical mistakes and run-on sentences.

Comment 8: Why only three groups of intestinal microbiota were analyzed?

Comment 9: Gut microbiota analysis generally provides diversity index, structure and composition of the bacterial communities, at the phylum and genus level, differential microbiota. Functional predictions are rarely used.

Comment 10: How to treat Chlorella vulgaris price higher than soybean meal?

Reviewer 2 Report

Review Manuscript ID: animals-2497812

This work aimed to study the effect of replacing soybean meal protein by microalgae Chlorella vulgaris powder on the growth and intestinal health of grass carp. It studies diverse physiological features in fish. Authors found that Chlorella vulgaris powder replacement of 50% soybean meal can be recommended as feed for carp. This study is an important contribution to finding novel protein sources to keep up with the sustainability of the aquaculture sector. This manuscript is generally well written, with minor concerns being addressed. After correcting these minor traits, the present manuscript can be accepted for publication. 

Minor revisions

- Line 24: where it is “trail” should be “trial.”

- The sentence starting in line 25 should be more straightforward: “The results indicate that the highest final body weight (FW), weight gain rate (WGR) and protein efficiency ratio (PER) were observed in the X50 group (p < 0.05).” In line 25, authors stated that the X50 group exhibited the highest FW, WGR and PER and in the sentence starting in line 27, authors state that the X50 group was lower than the control group. What parameter is significantly lower? Please retype this sentence to make it clearer.

- Line 63: delete “was more” and replace it by “is”.

- Line 72: replace “is limited until now” by “is still limited”.

- Line 87: replace “die” by “diet”.

- Line 111: Sentence should be rephrased to make it clear. Authors do not mention that the experimental diets were tested in triplicate. This information must be indicated. Suggestion: “750 grass carp were randomly assigned into 5 groups in triplicate.”

- Line 125: authors state that midgut samples were put in 4% paraformaldehyde and in line 148, state that samples were fixed with 10% neutral buffered formalin. To my knowledge, these two histology fixatives are the same, therefore this information should be coherent in the manuscript.

- Line 150: authors should indicate the meaning of “HE” initials in the text.

- Line 190: Authors say that results were represented by the mean ± SD but in the legend of figure 2 (line 256), gene expression results are expressed as mean ± SE. Please, correct this information in the statistical analysis section.

- Line 191: Confusing sentence. Should be rephrase it. Suggestion: The Duncan’s multiple range test was used to determine the significant differences between experimental groups when p < 0.05.”.

- Line 195: The title of this subsection should be improved. Suggestion: “3.1. Feed utilization, indices, growth and biometric parameters”.

- Line 196: This sentence should be deleted or rephrased since suggest a timeframe sampling when authors refer that FW, WGR and PER increased first and then decreased, which is not the case in this study. Authors should be more objective. Suggestion: “At the end of the experimental trial the X50 group showed higher FW, WGR and PER than the SM group (p < 0.05)”.

- Instead of indicating that p < 0.05 or p > 0.05 in the text, authors should indicate the exact value of p-value since it will inform the readers how much is near or far from α = 0.05. This adjustment will benefit the manuscript quality.

- Line 198: Rephrase this sentence to make it clearer. Suggestion: “The fish from the experimental X50 group exhibited a lower FCR value when compared to the one observed for the control group.”

- Be careful with using the word “significant” as a synonym for statistically different. This is repeated several times throughout the manuscript.

- Line 202, line 206 and line 235: “…among the 5 experimental groups.”

- Line 228: This sentence is confusing. Suggestion: “In contrast, a markedly reduced injury of middle intestinal villi were observed in the X50 group.”

- Line 237, line 239/240 and line 266: “…significantly up-regulated in…”. Please verify if the manuscript displays more of this trait.

- Line 258 to 260: The same as stated for line 196. Authors should be more objective. This sentence should be deleted, starting this section by: “The highest SOD value was observed in the X50 group when compared with the control group.”

- Be careful when using the words “control group” instead of SM group. Make sure that this correspondence is referred in the text to avoid misinterpretations.

- Line 260: Authors should rephrase this sentence to become clearer. Suggestion: “The CAT activity decreased in the group of fish fed diets with more than 50% soybean replacement by C. vulgaris powder, when compared with the SM group.”

- Line 266: “… down-regulated…”. Please verify if the manuscript presents more of this trait.

- Line 268: replace “had” by “showed”.

- Line 269: This sentence is incomplete. There was no significant differences between the SM group and X100 group concerning what?

- Be careful when describing the results and ask yourself and others if the sentence is clear. This misreading is repeated several times throughout the manuscript.

- Line 281: The same as stated previously. Authors should be more objective. This sentence should be deleted, starting this section by: “The AKP, ACP and LZM activities were higher in the X50 group compared to the SM group.” 

- Line 432: “Inflammatory response regulates the immune system by…”

- Why “human body” and “conditioned pathogen” words are underlined in line 456 and 464, respectively?

- Line 480: “…with high level of C. vulgaris powder.”

- If authors do not use numbered references in the text, it is not worth it to number them in the references section.

English quality is adequate for understanding the manuscript, therefore for publication.
